# The development of autonomous unmanned aircraft systems for mosquito control

Gregory M. Williams[ID][1,2]*, Yi Wang[2], Devi S. Suman[3], Isik Unlu[2], Randy Gaugler[2]

**1** Hudson Regional Health Commission, Secaucus, New Jersey, United States of America, **2** Center for Vector Biology, Rutgers University, New Brunswick, New Jersey, United States of America, **3** Zoological Survey of India, New Alipore, Kolkata, India

* gwilliams@hudsonregionalhealth.org

**Data Availability Statement:** All relevant data are within the manuscript and its Supporting Information files.

**Funding:** This work was supported by the USDA National Institute of Food and Agriculture Hatch

## Abstract

We constructed an electric multi-rotor autonomous unmanned aerial system (UAS) to perform mosquito control activities. The UAS can be equipped with any of four modules for spraying larvicides, dropping larvicide tablets, spreading larvicide granules, and ultra-low volume spraying of adulticides. The larvicide module sprayed 124 μm drops at 591 mL/min over a 14 m swath for a total application rate of 1.6 L/ha. The tablet module was able to repeatedly deliver 40-gram larvicide tablets within 1.1 m of the target site. The granular spreader covered a 6 m swath and treated 0.76 ha in 13 min at an average rate of 1.8 kg/ha. The adulticide module produced 16 μm drops with an average deposition of 2.6 drops/mm$^2$. UAS pesticide applications were made at rates prescribed for conventional aircraft, limited only by the payload capacity and flight time. Despite those limitations, this system can deliver pesticides with much greater precision than conventional aircraft, potentially reducing pesticide use. In smaller, congested environments or in programs with limited resources, UAS may be a preferable alternative to conventional aircraft.

## Introduction

In 1972, a team from the University of Delaware constructed and flew a radio controlled (RC) plane equipped with a miniaturized ultra-low volume (ULV) spray system [1]. While this marked the first time a remotely piloted aircraft was fitted with an insecticide spray system, it was developed solely for studying spray drift. Eight years later another team from the University of Delaware used a large RC airplane to apply the mosquito adulticides dibrom and malathion to a salt marsh to study the toxicity to killifish [2]. The first serious attempt to use remotely piloted aircraft for insect management occurred just a few years later when scientists from the US Department of Agriculture utilized an RC biplane with an eight-foot wingspan to control fall webworms and walnut caterpillars [3]. While the results were comparable to conventional aircraft, the technology remained impractical, noting "piloting a plane this size from the ground with any precision is more complicated than flying a real airplane" [4].

In 1980, Japanese researchers initiated a program to develop remotely piloted helicopters for agricultural spraying [5]. This program eventually led to the development of the gasoline-

project accession number 1020755 through the New Jersey Agricultural Experiment Station, Hatch project NJ08530, the Rutgers Center for Unmanned Aircraft and the Northeastern Mosquito Control Association's McColgan Grant-in-Aid. The funders had no role in study design, data collection and analysis, decision to publish, or preparation of the manuscript.

**Competing interests:** The authors have declared that no competing interests exist.

powered RMAX® (Yamaha® Motor Co., Ltd., Shizuoka-ken, Japan), an RC helicopter purpose-built for spraying rice fields. The Department of Defense explored the use of the RMAX for mosquito control spraying to protect soldiers overseas from biting insects [6]. Unfortunately, the RMAX was expensive ($86,000 - $1,000,000 US depending on configuration) and required extensive training to fly [7]. Those factors, combined with prohibitive regulations and technical difficulties, caused the military to transfer the project to the US Department of Agriculture in 2004 [8]. Researchers made significant improvements to the application technologies [9, 10] but unmanned aircraft systems (UAS) remained too expensive and difficult to fly to be practical for mosquito management.

In 2010 the AR Drone® (Parrot® SA, Parris, France) was released. This was the first mass-produced multirotor aircraft aimed at the consumer market. The AR Drone's electric power and advanced onboard sensors made the machine relatively easy to fly without specialized RC flight training. This single product was revolutionary, launching a frenzy of multirotor aircraft development and more importantly advanced flight control systems. That rudimentary toy demonstrated the immense potential of unmanned aircraft to mosquito programs and launched our research and development of UAS for mosquito management.

Conventional aircraft require large areas to operate effectively and the low altitude maneuvers required by many mosquito management applications are difficult in congested environments. Urbanization around wetlands, power lines, cell phone towers, and wind turbines make it increasingly difficult for aircraft to apply pesticides to these areas. Unmanned aircraft can fly at very low altitudes, can use sonar, radar or laser sensors to follow changes in terrain altitude. Some come equipped with obstacle detection sensors which prevent collisions and permit autonomous maneuvers around obstacles. In smaller, fragmented environments, UAS may be preferable to conventional aircraft. Cost has become an important issue for public agencies in recent years as budgets have shrunk or stagnated [11]. For mosquito management programs with limited budgets, a small fleet of UAS would cost the equivalent of one day of contracted aerial application service, making aerial applications available to any size mosquito control program in any type of environment.

Nearly four decades have passed since those first flights over the wetlands of Delaware and the technology has advanced exponentially. Today, UAS are in common use in agriculture [12], but adoption by professional mosquito control programs has lagged [13]. This project was conducted to develop novel aerial application technology by constructing interchangeable modules for dispensing mosquito control products and to determine the feasibility of UAS in comparison to conventional aircraft. Utilizing a commercial heavy lift multirotor aircraft, we developed several modular systems for conducting mosquito management activities. Each module was developed to be completely autonomous so that all flights could be programmed and executed by novice pilots with no additional input from the user during a mission. Swappable modules were constructed for larval surveillance, larval collection, adult collection, larval control with liquid and solid products, and adult control. We describe the design and testing of our larval and adult control modules as well as perspectives on the future development of UAS in mosquito management.

## Materials and methods

### Aircraft

A Spreading Wings S1000+® octocopter (DJI® Science and Technology Co., Ltd., Shenzhen, China) was the UAS platform selected for development (Fig 1A). The UAS was equipped with a Pixhawk® (3D Robotics®, Inc., Berkeley, CA) flight control system (FCS) because it offered flexibility in programming and controlling peripheral devices. The FCS included a global

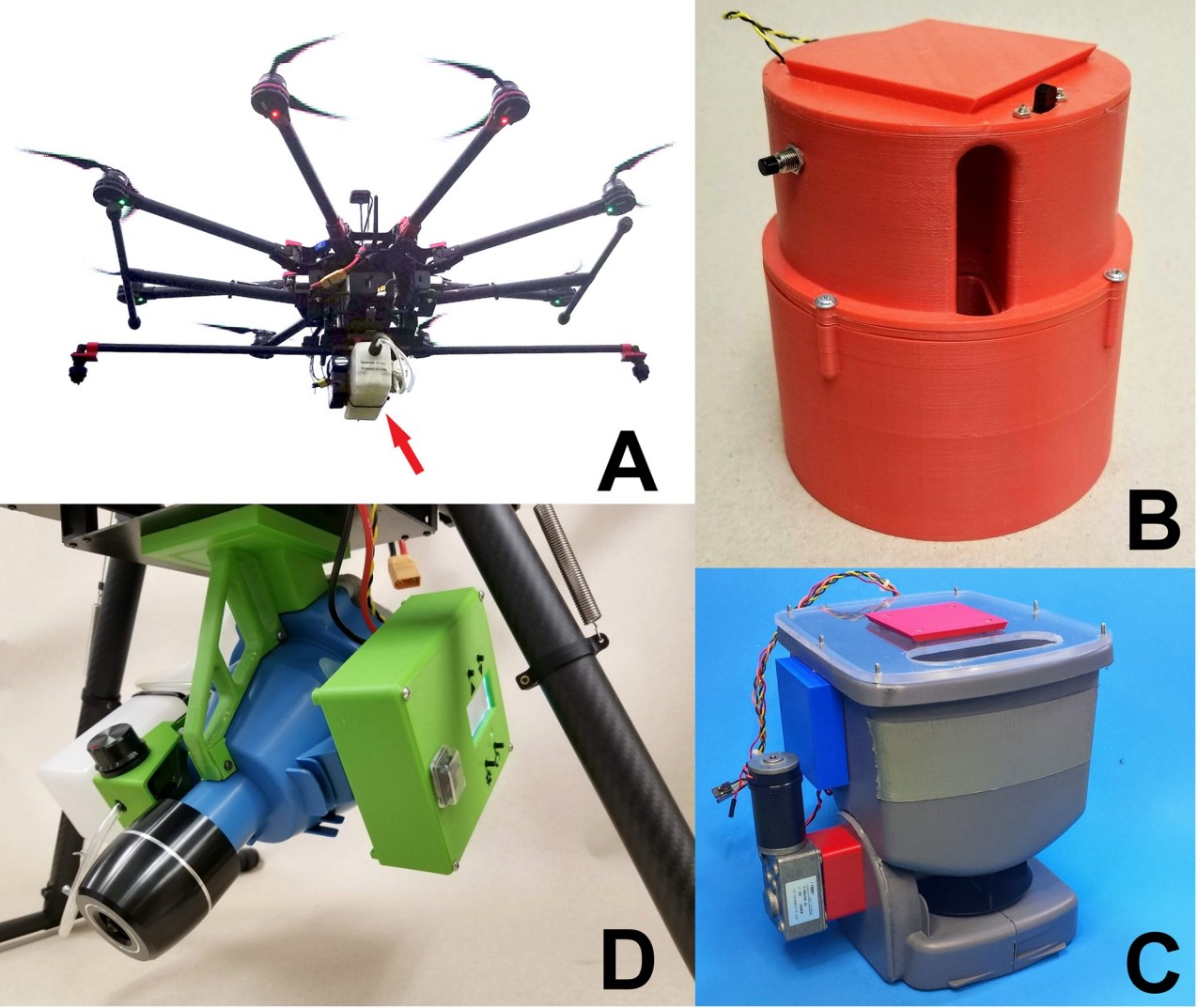

**Fig 1.** A. DJI S1000+ UAS and spray module (arrow) for dispensing liquid larvicides. B. Dispenser module for dropping Natular XRT tablets from a UAS. C. Modified hand spreader module for the application of larvicidal granules from a UAS. D. Longray ultra-low volume spray module for mosquito adulticide application from a UAS.

positioning system (GPS) receiver for navigation. A custom 3D-printed acrylonitrile butadiene styrene (ABS) plastic quick-detach dovetail shoe was affixed to the bottom of the UAS for mounting accessory modules such as spray systems. Accessory modules were controlled using digital or analog signals from the FCS through the auxiliary output pins.

Manual flights were conducted with a Taranis® 2.4 GHz radio transmitter and receiver (FrSky® Electronic Co., Ltd., Jiangsu, China) allowing a range of over 1.5 km. Autonomous flight was accomplished by programming missions through a ground station consisting of Mission Planner® flight control software [14] on a personal computer or the QGroundControl® application [15] on a smart phone or tablet. The complete system cost approximately $5,800 US with two batteries, but not including the cost of a smartphone, tablet, or laptop computer for the ground station. The price of battery charging systems varied greatly but will add $100 to $500 US to the overall cost.

## Insecticide sprayer

A spray system (Fig 1A) was constructed to dispense liquid larvicides by adapting the design of Huang et al. [10]. A 1-L polyethylene tank held the spray material. The liquid was pumped with a 12V miniature gear pump (EW-07012-20, Cole-Parmer®, Vernon Hills, IL) through 9.5 mm vinyl tubing to a pair of QJ100 series Quick TeeJet® nozzles (Spraying Systems® Co., Wheaton, IL). The nozzles were attached to the ends of 1.2 x 106 cm carbon fiber tube that extended to either side of the UAS. The supply tubing was routed through the boom. Flow rate was controlled by a motor controller (18v7, Pololu® Corp., Las Vegas, NV) which used pulse width modulation to control pump speed. The motor controller was connected to the FCS through an auxiliary output pin which was configured through the flight control software to control the speed of the spray pump. Flow rate could be configured to be constant or proportionately linked to the speed of the aircraft. All components were mounted to a carbon fiber plate with a 3D-printed ABS dovetail that slid into the octocopter shoe. The pump produced a maximum pressure of 2 bar, flowed up to 2 L/min and could run continuously for a minimum of 45 min on a 3 cell 2,200 mAh rechargeable LiPo battery. Empty weight of the spray system was 0.71 kg. The nozzles were compatible with most TeeJet spray tips, permitting a multiplicity of flow rates, spray patterns and droplet sizes simply by swapping the nozzles. The spray system cost under $200 US to construct.

Eight TeeJet XR extended range flat fan nozzles (Spraying Systems Co., Wheaton, IL) were selected for evaluation. Maximum flow rates were determined by running the pump at 12 v and discharging the nozzles into a container for 1 min with either BVA 13® oil (Trident Industrial®, New Hudson, MI) or water. The liquid was collected and the volume that was dispensed was measured using a precision balance (model PX523, Ohaus Corp., Parsippany, NJ) (n = 3 replicates each).

Based on the flow rate tests, XR11001 nozzles were selected for further testing. Aerial applications were made over an open field to establish application rates and droplet sizes. Twenty-one 5.08 x 7.62 cm Kromekote® spray cards (Smart Papers®, Hamilton, OH) were staked 1 m above the ground and 1 m apart in a 20 m line. Three card lines were set out perpendicular to the wind with 9 m between each line for a total of 63 cards per test. The sprayer was calibrated to dispense BVA 13 oil at 591 mL/min. The oil was mixed with FD&C Red 40 Granular DM food dye (Glanbia Nutritionals®, Carlsbad, CA) at a ratio of 20 g/L so that droplets were visible on the cards. The UAS was programmed to take off 50 m downwind of the first card line, activate the sprayer, make one application pass over the center of the plot at an altitude of 10 m at 4.4 m/sec, continue 50 m beyond the last card line, shut off the sprayer, return to the launch site, and land. The entire mission was conducted autonomously; the only input from the pilot was to start the mission. Cards were collected 10 min post-flight and replaced for a total of three replicates. The cards were scanned and analyzed using the DropVision® Ag image analysis software (Leading Edge Associates®, Waynesville, NC) to measure effective swath width and droplet sizes. Droplet spectra were described as the Sauter mean diameter, $D_{V0.1}$, $D_{V0.5}$ and $D_{V0.9}$, which define the proportion of spray volume (10%, 50% and 90%, respectively) contained in droplets of the specified size or less.

## Tablet dropper

We constructed a dropper module (Fig 1B) to dispense solid larvicide tablets (Natular® XRT, Clarke Mosquito Control Products, Roselle, IL). The module was 3D-printed in ABS and consists of an outer shell and an inner carousel holding eight tablets, weighing 410 g empty and 730 g full. Tablets were loaded through a port in the top of the housing. To release a tablet, a servomechanism (Hitec® RCD USA, Inc., Powa, CA) advanced a ratchet-and-pawl

mechanism to rotate the carousel one position, moving a tablet over an opening in the base of the housing. A Servo Trigger® (Sparkfun® Electronics, Niwot, CO) controlled the motion of the servomechanism and was activated by the FCS which output a 3.3V relay signal. The system was powered by a 2 cell 1,500 mAh rechargeable LiPo battery regulated to 6V which was sufficient for over 8 hrs of run time. The module was attached by a dovetail that mates with a shoe on the UAS. The module could autonomously drop a tablet via the flight control software by setting the output pin to trigger the relay signal at predetermined locations or it could be manually triggered with the radio transmitter. The construction cost was approximately $75 US.

The accuracy of the UAS's ability to autonomously drop tablets was tested over a 15 x 30 m plot. The UAS was centered on a marker at each of the four corners to record accurate GPS locations in the flight control software. The UAS was programed to autonomously take off, fly at 2.2 m/sec and 4 m altitude, drop a tablet at each of the four target locations and then return to the launch site and land. The aircraft paused for five seconds at each drop site to stabilize. As the tablets were dropped, a spotter immediately placed flags where the tablets impacted the ground to differentiate the impact point from any bounce of the tablets. The distance and direction between each corner marker and the impact point was recorded (n = 3 replicates). Wind speed and direction were recorded with a weather meter (model 2500, Kestrel Instruments®, Boothwyn, PA) to measure any influence on the tablet trajectories.

## Granular spreader

We developed a module to distribute larvicidal granules by modifying a hand spreader (LB6306, Vigoro® Corp., Lake Forest, IL) (Fig 1C). The hand crank which turns the impeller was replaced with a 12V, 160 RPM right angle gear motor (DongGuan Tsiny Motor Industrial Co.®, Guangdong, China) using a 3D-printed ABS adapter to mate the motor shaft to the impeller gear for a final impeller speed of 617 rpm. Motor speed was regulated by a motor controller (Pololu Corp., Las Vegas, NV), and a servomechanism (Hitec RCD USA, Inc., Powa, CA) was used to actuate the trigger controlling the hopper flow gate. The motor controller and servomechanism were plugged into separate ports on the FCS so that impeller speed and hopper flow could be controlled independently. The flight control software could regulate the impeller speed and hopper flow to maintain a constant application rate based on the speed of the aircraft. A 3-cell 460 mAh LiPo battery provided direct power to the motor and 5V to the servomechanism through a voltage regulator with enough capacity to run the spreader for over an hour. The lid was constructed of clear acrylic to view the hopper contents and provided a location to attach the dovetail mount. The module had an empty weight of 1.2 kg. The hopper had a volume of 2.9 L and could hold 1.1 kg VectoBac® G (Valent Biosciences® Corp., Libertyville, IL), 1.7 kg VectoLex® FG (Valent Biosciences Corp., Libertyville, IL), or 2.8 kg Altosid® Pellets (Wellmark® International, Schaumburg, IL), enough to treat 0.4, 0.3, and 1 ha respectively at the minimum application rates. The cost of constructing the granule spreader module was approximately $145 US.

Static flow rates were calculated for VectoBac G (5/8 mesh) and VectoLex FG (10/14 mesh) by activating the spreader on the ground for 30 sec and collecting the granules in a bucket with the hopper gate set at 4, 10, and 20 mm open. The larger Altosid pellets were tested with the gate at 7, 10, and 20 mm open. After each collection, the granules were weighed to calculate the flow rate (n = 3 replicates each). To establish optimal application altitude, the UAS hovered at altitudes of 3, 6, and 9 m and VectoBac G and VectoLex FG were dispensed individually at a rate of 136 g/min. An observer marked the swath extremes with flags while the granules were being dispensed to measure the swath width and offset from the UAS. Once the optimal

altitude was determined, aerial swath width and application rate tests were conducted with VectoPrime® FG (Valent Biosciences Corp., Libertyville, IL) at 136 g/min. A line of 68-L plastic containers was set up in a field perpendicular to the prevailing winds. The containers had an opening of 0.42 x 0.57 m and were spaced 0.6 m apart over 9.8 m for a total of 16 containers in the row. The UAS was programmed to autonomously take off 15 m downwind of the container line, activate the spreader, make one application pass over the center of the line at 3.3 m/sec and an altitude of 6 m, continue 15 m beyond the line, shut off the spreader, return to the launch site, and land. After each flight, the granules in each container were collected, weighed with a precision balance (model PX523, Ohaus Corp., Parsippany, NJ) and used to calculate the swath width.

### Adulticide ULV sprayer

We modified an electric handheld ULV sprayer (Shenzhen Longray® Technology Co., Ltd., Shenzhen, China) to control adult mosquitoes, (Fig 1D). The sprayer used pressurized air to atomize liquids as they exited the nozzle. Air from the fan pressurized the tank and forced liquid to the nozzle. The sprayer was disassembled, and the housing and spray tank were discarded to reduce weight to 1.75 kg. New housings and mounts were 3D-printed in ABS plastic and the 2.5-L tank was replaced with a lighter 600 mL tank. The module received power from the flight battery and was activated by the FCS which sent a 3.3V signal to the sprayer activation button. Blower speed was 18 m/sec and the flow rates ranged from 40 to 325 mL/min. Flow rates were adjusted manually, and flight speeds were matched to the flow rate during spraying for accurate application rates. Droplet sizes were directly proportional to flow rate and ranged from 23 to 58 μm. The original cost of the ULV spray unit was $1000 US with an additional $20 US in parts for the housing, tank, and shut-off valve. Laboratory calibration of the sprayer was performed at a flow rate of 50 mL/min with the Army Insecticide Measuring System (AIMS) [16].

For aerial ULV trials, nine rotating impactors [17] were staked 1.5 m above the ground in a 3 x 3 square pattern with 15 m separating each. Impactors were activated before each application and stopped 10 min after the application to collect aerosolized droplets on two 3 x 3 x 75 mm acrylic rods coated with Teflon® tape. BVA 13 oil marked with a fluorescent tracer dye (Tinopal® OB, Ciba Corporation, Newport, DE) was applied at a rate of 50 mL/min. The UAS was programmed to autonomously take off 30 m downwind of the plot, turn on the sprayer, and fly a back and forth pattern over the plot (6 m altitude at 3.3 m/sec) with 6 m between each flight line. The flight paths were extended 10 m beyond the plot in all directions to ensure total coverage. Following the application, the UAS shut off the sprayer, returned home, and landed autonomously (n = 3 replicates). All droplet diameter measurements and density calculations were performed under a fluorescent compound microscope using DropVision FL image analysis software (Leading Edge Associates, Waynesville, NC).

### Statistics and modeling

All statistical analyses were performed in SAS® Studiosoftware, version 3.8 of the SAS system for Windows [18]. Means and standard errors were calculated using the Summary Statistics task and the Linear Regression task was used to calculate r-square values. Computer-aided design was conducted in Fusion 360™ version 2.0 for Windows (Autodesk®, San Rafael, CA). Three dimensional models were prepared for printing in Ultimaker Cura software and printed on an Ultimaker 2+ (Ultimaker North America, Waltham, MA). All STL models are available for download (S1–S4 Files).

## Results & discussion

### Insecticide sprayer

At maximum power (12 v), laboratory flow rates for nozzle pairs ranged from 620 to 2040 mL/min with water and 590 to 1740 mL/min with BVA 13 oil (Fig 2A). Analysis of spray cards

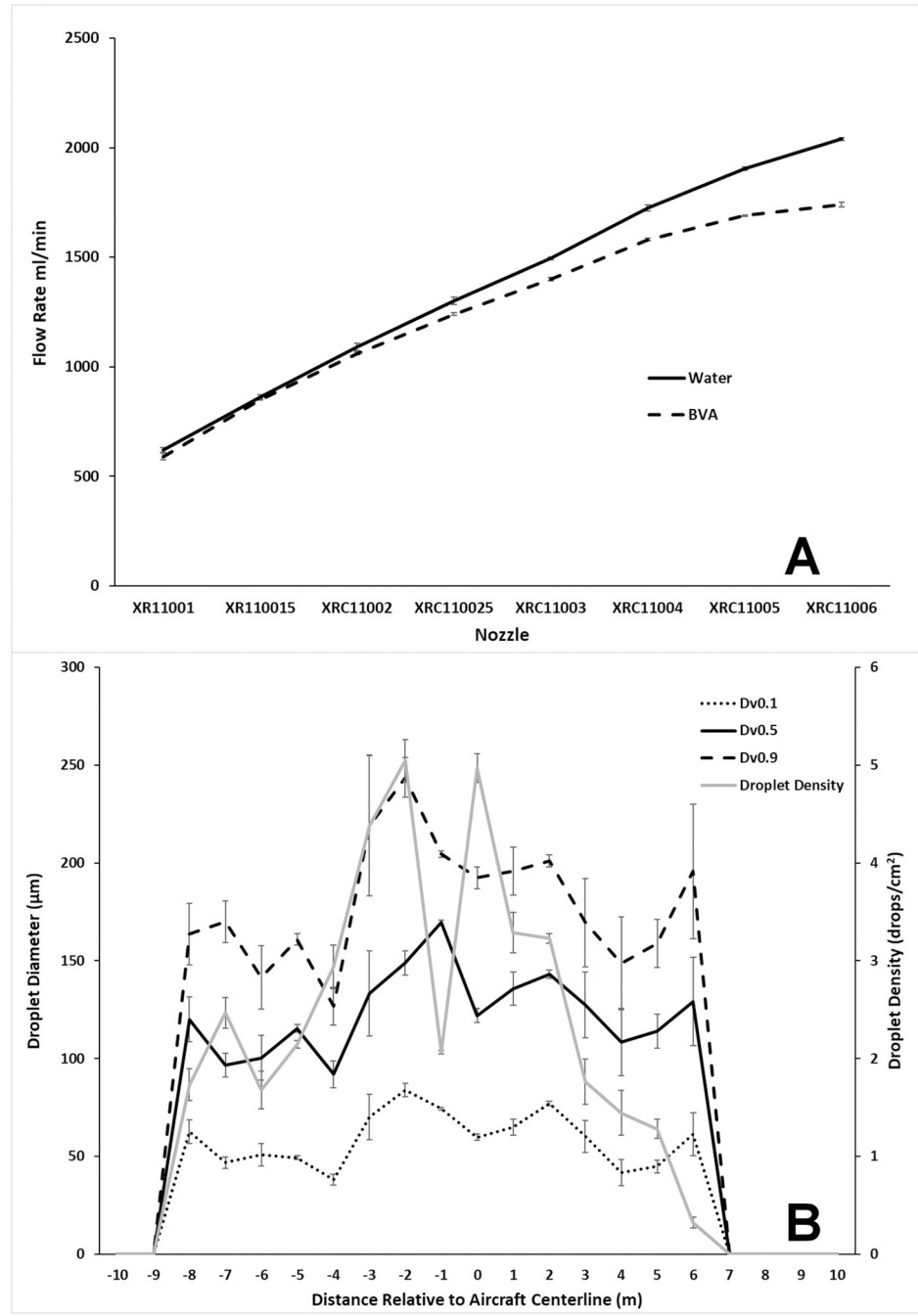

**Fig 2.** A. Average flow rate for two XR TeeJet nozzles applying water or BVA 13 oil at a pressure of 2 bar ± SEM. B. Swath width and droplet diameter (left axis) and droplet density (right axis) of BVA 13 oil (± SEM) applied at a rate of 591 mL/min from DJI s1000+ UAS flying 4.4 m/sec at an altitude of 10 m.

confirmed median droplet sizes ranging from 91.90–169.48 μm across a maximum swath of 14 m with a slight propensity for larger droplets closer to the aircraft flight line (Fig 2B). Median droplet size across all replicates was 123.68 μm ($D_{V0.1}$ = 59.02 μm, $D_{V0.9}$ = 179.42 μm). Droplet density averaged 2.58 drops/cm$^2$ (range = 0.32–5.05 drops/cm$^2$) with decreased density directly below the aircraft likely due to propeller vortices forcing drops away from the aircraft (Fig 2B). The flow rate of 591 mL/min, 4.4 m/sec flight speed and a 14 m swath resulted in a final application rate of 1.6 L/ha; a rate suitable for high application rate larvicides such as VectoBac 12AS (label rate: 0.29–2.34 L/ha). Based on our calculations, VectoBac WDG (label rate: 2.4–93.5 L/ha) can also be applied if flight speed is reduced to 3 m/sec. Reducing the flow rate to 107 mL/min by lowering the pump voltage would result in an application rate of 0.29 L/ha, appropriate for low application rate larvicides such as Natular 2EC (label rate: 0.08–0.46 L/ha) or Altosid SR5 (label rate: 0.22–0.29 L/ha). Based on our flight tests, the UAS can fly for approximately 23 minutes with the spray system attached. With a 14 m swath and flying at 4.4 m/sec, this system should cover 8.5 ha in a single battery.

The Huang et al. [10] system employed four rotary atomizers to deliver small droplets (37–66 μm), whereas we deployed two flat fan nozzles delivering much larger 124 μm drops. There is a wider variety of flat fan nozzles which are lighter and use less power than the rotary atomizers. Huang et al. [9] recovered material up to 42 m downwind of the release point with their system while our tests yielded a swath of 14 m. This discrepancy is likely due to the more sensitive collection method used and the fact that the spray system was run from a pole mounted on a truck, not from a running helicopter. The vortices created by the helicopter will push material down, reducing the total swath [19]. Finally, the smaller droplets of their system will drift farther [20]. To produce smaller droplets, we have tested the system with 203 μm misting nozzles (AeroMist$^{®}$, Inc. Phoenix, AZ) which produced $D_{V0.5}$ 43–65 μm droplets depending on altitude at a flow rate of 55 mL/min. The advantage of the larger nozzles is that our system pumped nearly six times the amount of material compared to Huang et al. [9] making it compatible with higher application rate larvicides and faster application speeds.

Kimball and Reynolds [21] also used rotary atomizers for their larvicide system. Although swath and droplet deposition were not reported, they achieved 99.2% mortality of caged *Culex tarsalis* Coquillett in an open water site using a tank mix of VectoBac and VectoLex WDG at 0.29 and 0.88 L/ha, respectively. While we did not conduct bioassays, the droplet deposition data were comparable to droplet and bioassay results from truck-mounted sprayers [22] which indicate that our system can distribute larvicides at the proper application rate for common mosquito larvicides and should result in excellent efficacy.

## Tablet dropper

The tablets landed an average of 1.1, 95% CI [0.93, 1.35] from the target site (Fig 3). At each site, the tablets landed in the same general location demonstrating good repeatability (range = 0.66–1.38 m). Due to the low altitude of the UAS and 40 g weight of the tablets, the gusting 1.6–3.4 m/sec wind had no effect on accuracy (average distance of group from the target) as most tablets landed northeast of the target despite a northerly wind. Errors can be attributed to the resolution of the GPS unit. Overall, accuracy and precision (farthest distance between replicates at a site) increased as the mission progressed with the smallest and closest group occurring at the last drop location ($R^2$ = 0.94 & 0.87 respectively).

Accuracy relied on the GPS module and the ability of the UAS to fly to the correct location and hold position under varying environmental conditions. Overall, the UAS performed extremely well during the autonomous tests. While the manufacturer's reported accuracy of the GPS module was 2.5 m [23], the UAS was able to repeatedly drop tablets within 1.1 m of

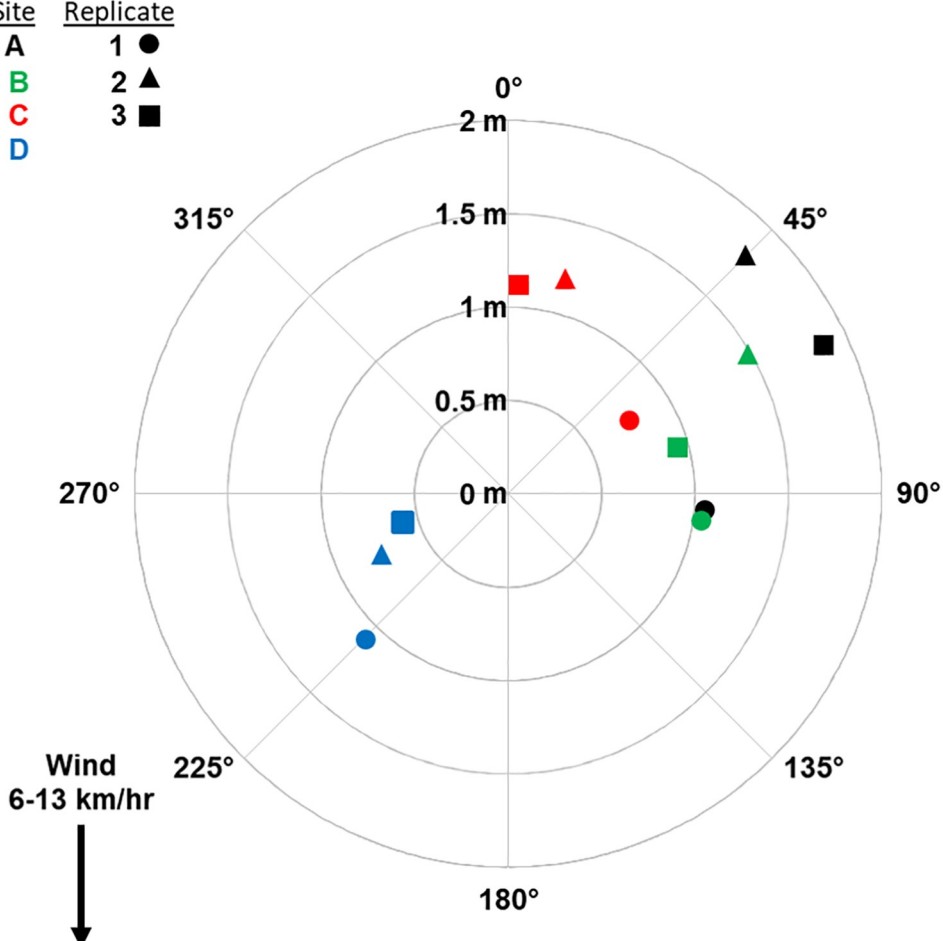

**Fig 3. Placement accuracy of Natular XRT tablets autonomously dropped from a UAS.** Distance (m) and position (degrees) of each tablet in relation to intended center target (0,0). Target sites are represented by color and replicates are designated by shape.

the target. Accuracy improved as flights progressed because the UAS had more time to collect and correct for GPS data as indicated by the number of GPS satellites available and signal strength data recoded by the flight controller (S1 Fig). With a maximum distance of 2.9 m between all drops, the UAS can accurately deliver a tablet to any target at least that radius. In our UAS, the GPS unit only received signals from the US GPS system, but accuracy can be enhanced to < 1 m by using newer navigation modules having access to multiple satellite networks (GLONASS, Galileo, BeiDou, etc.) [24]. Further, adding a second ground-based GPS unit for real time kinematic (RTK) navigation would increase accuracy to several centimeters [25].

The tablet dropper was designed around the Natular XRT tablet because the heavy mass and streamlined shape helped it fall straight down when released. However, the large size of the tablets limits the number that can be carried and requires a larger airframe. The tablet module can be easily redesigned and 3D-printed for any size, shape, or number of tablets.

This novel technology offers distinct advantages over the hand application of larvicide tablets in certain environments. At one of our research sites, hand application of tablets to 12 pockets of *Ae. sollicitans* on a *Spartina patens* marsh takes approximately 45 minutes not

including the 10-minute off-road drive to reach the site. The hike on the marsh is difficult and can only be accomplished during low tide, whereas the UAS can take off from a parking lot 900 m away, treat the sites under any tidal conditions, and return in under 9 minutes, an 80% improvement in efficiency. Newer, smaller tablet formulations such as Natular DT greatly increase payload capacity and would be ideal for micro sized UAS.

## Granule spreader

The static flow rate ranges for the spreader were (min-max ± STDDV): VectoBac G (2.58 ± 0.47–661.86 ± 56.65 g/min), VectoLex FG (5.28 ± 1.92–1,587.62 ± 81.49 g/min), and Altosid Pellets (42.35 ± 21.37–1,292.82 ± 43.57 g/min). The widest and most consistent distribution occurred at an altitude of 6 m (Fig 4). At 3 m altitude the swath was too narrow and at 9 m the larger, heavier VectoBac larvicidal granules had a swath of only 2 m. VectoPrime was selected for final testing because the low application rate of 1.4 kg/ha maximizes the treatment area per flight. Swath trials with VectoPrime yielded an average rate of 1.1 kg/ha over the 5.5 m swath with effective label rates (>1.4 kg/ha) occurring over a 1.8 m swath (Fig 5 –gray bars). When applying granules, the UAS flies over the plot, moves over for the next pass, turns 180 degrees, and flies over the next section in the opposite direction. The low application rates at the swath tails can be overlapped during each opposing pass to increase the application rate along the edges. Overlapping each application by 1.8 m (Fig 5 –blue bars) results in an average field application rate of 1.8 kg/ha, well within the recommended range for VectoPrime FG. With the existing hopper size, we can treat approximately 0.76 ha in 12.7 min at the minimum label rate before needing to reload the hopper. The flight battery can support at least two missions for a total application area of 1.52 ha per battery.

Previous studies have focused on applying liquid larvicides [9, 10, 21]. Most liquid larvicides are applied at a lower weight per area than granular products. This is especially important

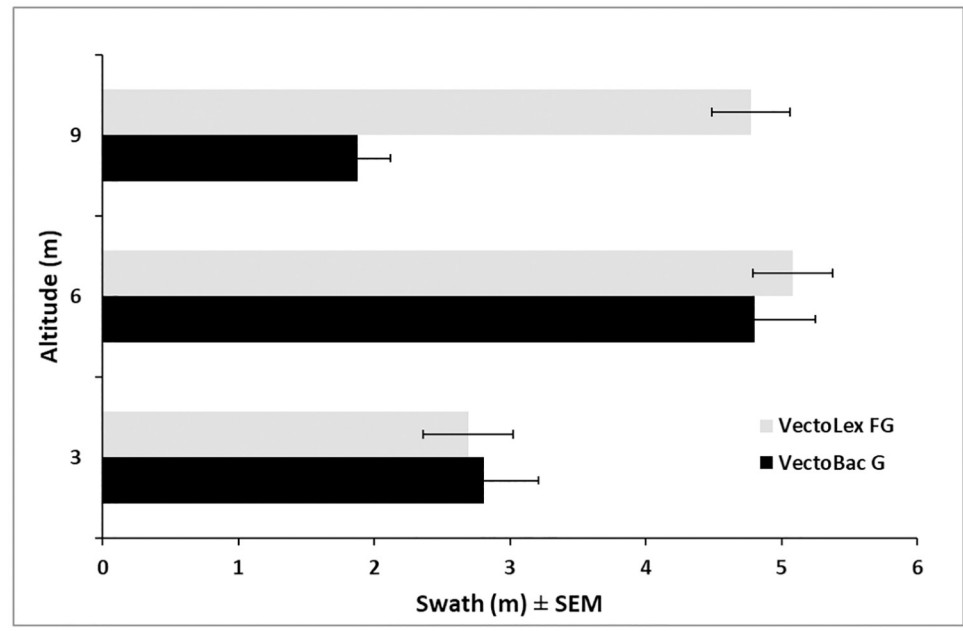

**Fig 4. Effect of altitude on swath width of VectoLex FG and VectoBac G larvicidal granules applied from a UAS ± SEM.**

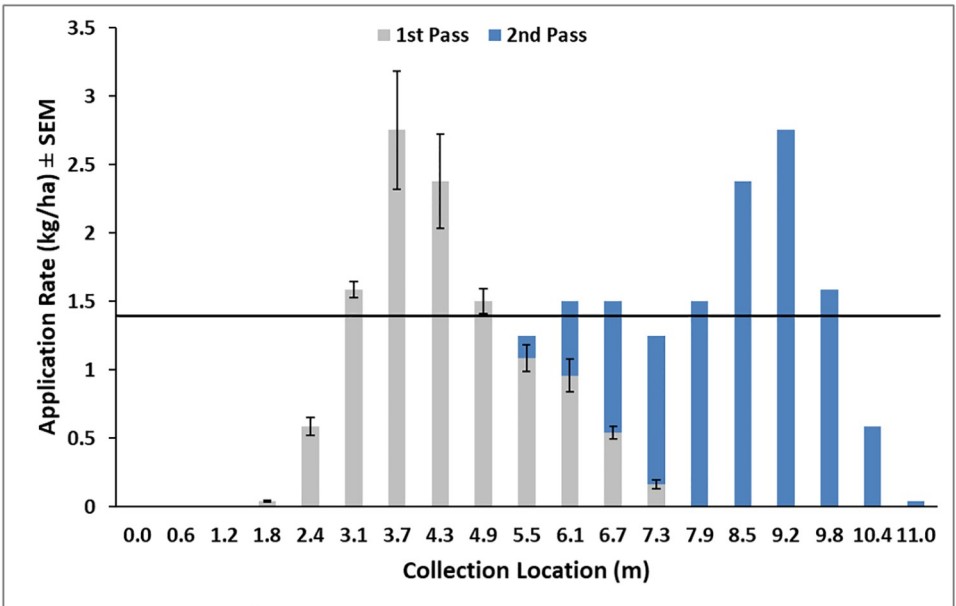

**Fig 5. Swath width and application rate test of VectoPrime FG applied at a rate of 136 g/min from a DJI S1000 + UAS at 6 m altitude over a line of containers ± SEM.** Each bar represents one container. Horizontal line signifies the minimum recommended application rate for VectoPrime of 1.4 kg/ha (maximum rate = 22.4 kg/ha). Gray bars illustrate one pass of the UAS. Blue bars illustrate a second pass in the opposite direction with a 1.8 m overlap. Average application rate for the plot = 1.8 kg/ha.

considering the limited flight times and payload capacities of currently available UAS. However, preliminary salt marsh field trials with VectoBac 12AS exhibited inconsistent larval mortality despite excellent droplet distribution presumably because much of the product was caught in the marsh grasses (S2 Fig). Based on conventional aerial applications, granular products penetrate vegetation better and yield greater mortality in coastal marsh habitats [26]. The US Air Force experimented with a granular applicator on the RMAX but determined that the flow rates were too high even at the lowest settings of the spreader, dispensing 16 kg of material in 20–30 min (533–800 g/min) [6]. Our system can flow VectoBac G at 2.6–661.9 g/min offering a greater range of application rates at various flight speeds. We encourage pesticide manufacturers to develop lighter granules with a higher concentration of active ingredient specifically for UAS.

## Adulticide ULV sprayer

Laboratory droplet tests with the AIMS produced a $D_{V0.5}$ of 22.74 μm (1,526 droplets counted), a $D_{V0.1}$ of 14.06 μm, and a $Dv_{0.9}$ of 119.25 μm. Fluorescent droplet analysis of the 54 acrylic rods measured an average of 3,629 drops with a $D_{V0.5}$ of 16.15 μm ($D_{V0.1}$ = 2.86 μm, $D_{V0.9}$ = 43.2 μm) and an average density of 2.6 drops/mm$^2$. The difference in median droplet size between the laboratory and field trials is likely due to the different measurement methods with the rotary impactor known to underestimate median droplet diameter [27]. The droplet diameter and density were consistent with mosquito adulticide label requirements and similar measurements have resulted in nearly 100% mortality in caged mosquito bioassays following ground-based ULV applications [28, 29]. Zhai et al. [30] applied deltamethrin from a UAS and achieved 100% mortality against caged mosquitoes with half the droplet density found in this study. The additional weight of the sprayer and power consumption of the blower reduced

flight time to 16 min. Given the same 6 m altitude and 6 m spacing between flight paths, these direct overhead applications can treat a maximum of 1.8 ha per battery. However, applying 31–50 μm droplets at an altitude of 15 m, Zhai et al. [30] achieved 100% mortality against caged mosquitoes with their UAS over a 137 m swath. Based on Stoke's law, a 150% increase in droplet size results in an 82% decrease in horizontal distance traveled [31]. Therefore, our smaller droplets should cover at least a 137 m swath. That would raise our maximum treatment area to 33 ha, limited by the amount of pesticide in the tank, not battery life.

The alternating back and forth flight pattern resulted in complete coverage across the plot with droplet densities ranging from 1.6–4.0 drops/mm$^2$. Truck-mounted applications are limited to navigable roads, occasionally far from the intended target site, and rely on the prevailing wind to carry the pesticide in the proper direction [32]. This poses a problem in densely populated cities where row home construction limits access to backyards and in rural areas where trucks are unable to reach the target site. Conventional aircraft can overcome these issues but have their own limitations. They are expensive to own or contract, have minimum treatment areas for economic reasons, often fly in dangerous low light conditions to coincide with peak mosquito activity and require carefully planned offsets to ensure the spray reaches the intended target [31]. Unmanned aircraft allow for adulticide applications directly to the target site which guarantees coverage, reduces the amount of pesticide required, minimizes drift, and eliminates the requirement for optimal wind conditions. They are also comparatively inexpensive and offer autonomous capabilities that reduce risk during low light operations. Unfortunately, current aerial application modeling software cannot account for the unique vortices created by multirotor UAS [19]. While upwind ULV applications are possible from UAS, more work is needed to determine optimal altitudes, droplet sizes, and offsets under various environmental conditions.

Ultra-low volume applications from a UAS pose a unique problem for pesticide labels. Adulticide labels typically categorize applications as ground-based or aerial. Equipment calibration for those two scenarios differ. Small droplets (< 30 μm) that will drift are preferred when applied from the ground, whereas larger droplets (<60 μm) that will fall with minimal drift and survive evaporation are preferred when applied from conventional aircraft [31]. UAS applications fall between the two extremes. Our applications were made at low altitude (6 m), therefore smaller droplets were appropriate. Higher altitude applications would require larger droplets but likely still smaller than conventional aircraft. More research is needed to establish optimal droplet sizes for UAS and in the future, we expect to see additional language on pesticide labels pertaining to UAS applications.

## Conclusions

The small size and limited flight time of most UAS translate into smaller areas treated compared to conventional aircraft. This is offset by the greater precision UAS offer. Conventional aircraft generally blanket mosquito habitat with larvicide even though larval distribution tends to be confined to smaller pools within the marsh [33, 34]. Relatively little product reaches the intended targets [35, 36]. Unmanned aircraft can precisely apply pesticides to much smaller areas than conventional aircraft, resulting in major reductions in insecticide use [37]. Further, in urban areas, buildings, cell phone towers and power lines create dangerous obstacles for conventional aircraft. UAS have little problem navigating in these crowded environments and in these areas, UAS may be the only option.

The combined payload capacity afforded through swarm technology will soon render the limited payload and limited swath of single UAS irrelevant. A swarm of 50 UAS flew in unison using the same flight control system we used in our octocopter [38]. That same technology

could be used to fly a swarm of multirotor aircraft, each with a small payload of pesticide, thereby achieving the application area of conventional aircraft. Landing pads with integrated battery charging capability already exist (Skysense, Inc., San Francisco, CA) and could be combined with automated pesticide filling stations [39] ensuring the swarms are always ready with little intervention. UAS can also be used for mosquito surveillance. Simple aerial images can identify sources of water while multi-spectral images can measure water quality and identify potential mosquito habitat [40, 41].

The risk involved with UAS operations is low. Although conventional aircraft are reliable, accidents do occur, sometimes with fatal results [42, 43]. With limited weight and speed, the risk of injury from UAS is low [44]. As UAS technology continues to improve, reliability and safety will increase further. Features such as obstacle avoidance (Skydio, Inc., Redwood City, CA) and redundant flight control systems (ZeroTech, Beijing, China; MicroPilot, Manitoba, Canada) are now standard features on many UAS. Geofencing creates virtual boundaries which prevent UAS from flying beyond a specified area or altitude. Loss of radio signal triggers the UAS to autonomously return home or land. Onboard aviation Automatic Dependent Surveillance-Broadcast (ADS-B) sensors and remote identification signals currently allow UAS to detect other aircraft [45].

When our project began, the only options for a UAS were to spend hundreds of thousands of dollars on a giant scale remote control helicopter, purchase a toy that could do little more than hover precariously in place or construct one using parts scavenged from hobby aircraft. Today, UAS are widely available with an estimated 6.4 million consumer units sold in 2015 alone [46]. Although most agricultural UAS are focused on aerial imaging and analysis, commercial manufacturers are developing UAS specifically designed for pesticide applications [47]. Less than ten years since the introduction of the first mass-produced consumer UAS, many mosquito control programs have embraced the technology. A recent survey of mosquito control programs found that 16% of respondents currently use UAS while 64% anticipated using UAS in the future [13]. We anticipate in the next several years, with the rapid advancement of UAS technology, most mosquito programs in the country will use UAS for some aspect of mosquito control, be it mapping, sampling, surveillance, or pesticide delivery. It is plausible that mosquito management will soon be conducted from a computer screen. Mosquito control professionals will trade in their dippers and boots for computers and virtual reality goggles. Already beginning in agriculture [48], teams of mosquito control UAS will soon work in unison. A simple keystroke will launch autonomous swarms of UAS. Sensor UAS will measure environmental conditions and map out potential habitat using advanced algorithms to predict mosquito populations, surveillance UAS will confirm mosquito activity, and control UAS will treat the infested microhabitats. Much of this work will happen at night when mosquitoes are most active, and people are indoors. After their work is complete, the UAS will land on special pads to automatically recharge the batteries and refill the pesticides, patiently awaiting their next mission. Just as the public once welcomed the sight of our spray trucks so will they welcome the sight of our UAS, knowing they will be just a little safer and more comfortable.

## Supporting information

**S1 File. STL files for UAS insecticide sprayer.**
(ZIP)

**S2 File. STL files for UAS tablet dropper.**
(ZIP)

**S3 File. STL files for UAS granule spreader.**
(ZIP)

**S4 File. STL files for UAS adulticide ULV sprayer.**
(ZIP)

**S1 Fig. GPS data logged by flight controller.** Total number of satellites broadcasting to GPS unit (NSats, top) and the horizontal dilution of precision (HDop, bottom). Lower HDop value indicates better GPS signal. Number of satellites increased and HDop decreased as flight time progressed.
(TIF)

**S2 Fig. Bioassay results of UAS application at 2.3 L/ha of VectoBac 12AS over a salt marsh.** Average (n = 10) 72 hr larval mortality in cups placed out in open and cups placed beneath partial cover of *Spartina patens* grass.
(TIF)

**S1 Data.**
(RAR)

## Acknowledgments

The authors thank Scott Crans for assistance conducting field trials, Shaun Kenny for help with engineering and troubleshooting, and Rafael Valentin for help designing early prototypes.

## Author Contributions

**Conceptualization:** Gregory M. Williams, Yi Wang, Randy Gaugler.

**Data curation:** Gregory M. Williams.

**Formal analysis:** Gregory M. Williams.

**Funding acquisition:** Gregory M. Williams, Randy Gaugler.

**Investigation:** Gregory M. Williams, Yi Wang, Devi S. Suman, Isik Unlu, Randy Gaugler.

**Methodology:** Gregory M. Williams, Yi Wang, Devi S. Suman, Isik Unlu, Randy Gaugler.

**Project administration:** Gregory M. Williams, Randy Gaugler.

**Resources:** Gregory M. Williams, Randy Gaugler.

**Supervision:** Gregory M. Williams, Randy Gaugler.

**Visualization:** Gregory M. Williams.

**Writing – original draft:** Gregory M. Williams.

**Writing – review & editing:** Gregory M. Williams, Yi Wang, Devi S. Suman, Isik Unlu, Randy Gaugler.

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
