## [Decision Letter · Decision Letter 0]

17 Mar 2020

PONE-D-20-02700

The development of autonomous unmanned aircraft systems for mosquito control

PLOS ONE

Dear Dr. Williams,

Thank you for submitting your manuscript to PLOS ONE. After careful consideration, we feel that it has merit but does not fully meet PLOS ONE’s publication criteria as it currently stands. Therefore, we invite you to submit a revised version of the manuscript that addresses the points raised during the review process.

We would appreciate receiving your revised manuscript by May 01 2020 11:59PM. To enhance the reproducibility of your results, we recommend that if applicable you deposit your laboratory protocols in protocols.io, where a protocol can be assigned its own identifier (DOI) such that it can be cited independently in the future. For instructions see: http://journals.plos.org/plosone/s/submission-guidelines#loc-laboratory-protocols

We look forward to receiving your revised manuscript.

Kind regards,

Ahmed Ibrahim Hasaballah

Academic Editor

PLOS ONE

Journal Requirements:

Please ensure that your manuscript meets PLOS ONE's style requirements, including those for file naming. The PLOS ONE style templates can be found at http://www.plosone.org/attachments/PLOSOne_formatting_sample_main_body.pdf and http://www.plosone.org/attachments/PLOSOne_formatting_sample_title_authors_affiliations.pdf

Reviewers' comments:

Reviewer's Responses to Questions

**Comments to the Author**

1. Is the manuscript technically sound, and do the data support the conclusions?

Reviewer #1: Partly

Reviewer #2: Yes

2. Has the statistical analysis been performed appropriately and rigorously? 

Reviewer #1: I Don't Know

Reviewer #2: Yes

3. Have the authors made all data underlying the findings in their manuscript fully available?

Reviewer #1: No

Reviewer #2: Yes

4. Is the manuscript presented in an intelligible fashion and written in standard English?

Reviewer #1: Yes

Reviewer #2: Yes

5. Review Comments to the Author

Reviewer #1: Summary and overall impression.

The authors of “The development of autonomous unmanned aircraft systems for mosquito control” describe a custom-built UAS and pesticide application modules for mosquito control. Under PLOS ONE Criteria for Publication, this manuscript may be considered a submission that describes a tool (i.e. an UAS that applies insecticides for mosquito control). This manuscript is focused upon and describes in sufficient detail the deposition of pesticides by the UAS and pesticide application modules (e.g. swath width, pesticide deposition rates, etc). Therein, they describe the construction and benchmark testing of a relatively low cost UAS that is fitted with custom modules that apply liquid, granular, tablet and adulticide pesticides for mosquito control. The module that deposits tablet forms of pesticide is highly innovative as to the best of my knowledge, no such device is commercially available. This aspect alone may be of great use to the mosquito control community, particularly for using UAS to apply insecticide inside of containers which support the growth of mosquitoes that spread diseases such as dengue. However, a more rigorous evaluation of the automated tablet dispensing module would have benefited the study (accuracy assessed with a total of 12 tablets dropped from the UAS). The data presented for each module attached to the UAS were within ranges reported previously for other aircraft used for mosquito control. An important advantage of using UAS to apply insecticide is that it can provide a more targeted application, potentially reducing the quantity of pesticide released into the environment. Although the quantity of pesticide deposited is within the range that should control mosquitoes, the manuscript does not provide data on whether the test applications made using their custom UAS actually were effective for controlling mosquitoes. In line 516 of the manuscript, they indicate that mosquito control data will be presented in a future publication. However, it is of some immediate concern that vector control agencies may elect to follow the manuscript methods and construct a low-cost UAS for mosquito control, without adequate data to show that it is indeed an effective tool. If the described UAS is not effective, others attempting to replicate it will waste public funds, and potentially put the public at increased risk for mosquito-borne diseases. This concern is highlighted by the authors in lines 394-396 where they state that applications of the liquid larvicide VectoBac 12AS from their UAS produced inconsistent larval mortality. While theoretical comparisons were made between the custom UAS and other aircraft used for mosquito control, data was not presented that demonstrate the custom UAS has a proven advantage over existing alternatives.

Major Issues

- Much of the manuscript is dedicated to describing how to build the UAS with pesticide application modules and a specification section describing the results of evaluations on pesticide deposition rates from the UAS. The study unfortunately lacks information on the effectiveness of the UAS for controlling mosquitoes, except in line 394-396 where they state that the larvicide that was applied from the UAS produced inconsistent larval mortality. To diminish concerns regarding the efficacy of this instrument, larval cup bioassays and caged sentinel adult mosquito bioassay should be conducted for applications from each of the pesticide application modules. Testing the UAS configured with most effective parameters (speed, swath width, etc.) in a homogenous setting (e.g. open field) with one insecticide for each module should not be laborious.

- The authors state in the submission questions that all relevant data are in the manuscript and Supporting Information files, but they do not provide the STL files that must have been used to program the 3D printer to make the pesticide application modules. Also not described is the software used to make the 3D models, or the 3D printer(s) that were used. These would be needed if other researchers attempt to replicate or build upon this study.

- The performance of the UAS and pesticide application system is compared to published articles and product manuals for other UAS used for pest control. Because environmental factors (e.g. wind, terrain, humidity) can strongly impact the efficacy of pesticide deposited from an aircraft, it would have been more compelling to compare the performance and efficacy of the UAS described in the manuscript with one that is commercially available and currently used in several industries (e.g. DJI Agras MG-1). A limitation of such comparison is that the Agras MG-1 has pesticide application modules for only liquid and granular products. However, application of tablets and adulticide by UAS is novel, so a direct comparison for these with another UAS may not be warranted.

- The Discussion near the end of the manuscript is overly lengthy and not as focused as it is earlier in the manuscript.

Minor Issues

- Lines 37-39. The statement that the UAS applications were similar to applications from conventional aircraft could lead readers to think that actual comparisons rather than calculations that were made. This should be clarified.

- Line 39: There is no data presented that pesticide use would be reduced with UAS applications, it is only inferred. Studies would need to be conducted to make this claim.

- Figures 1 – 3 do not present data, so could be included as Supplementary Information.

- Line 129: This reviewer searched the Cole-Palmer web site for “12V miniature gear pump” and received 44 results. Please provide the correct product number.

- Line 148: consider instead: “The liquid was collected and the volume that was dispensed measured using XXXX”; XXX being the instrument used to measure the volume.

- Line 260-61; 288: Dv0.1, Dv0.5 and Dv0.9 are not described in the methods.

- Line 263-64 The application rate of 0.09 L / ha is not suitable for Altosid SR-20 which the authors indicate is 0.07 L / ha. Curious why the authors would suggest the use of a product that could not be applied at the legal rate using the UAS

- Line 284: Please indicate which system produced 50 um and which 123 um.

- Line 298: The authors claim that the system “is able to produce excellent efficacy” without direct evidence is not fully supportive of that statement.

- The application rate of 0.09 L / ha appears to exceed the maximum application rate for aerial release of Altosid SR-20 (according to the product label).

- Lines 300 – 305: Authors should make it clear they are speculating that 6 ha could be treated with a single battery charge based on calculations.

- 305 – 306: Authors may wish to refrain from grandiose statements such as “the days of blanketing an area with pesticide are numbered.” It is overly broad (area is not defined) and unlikely to be the case in the near to distant future as large areas such as marsh habitats, rice fields, or pastures are still likely to be braodly treated for mosquito larvae.

- Line 308: Please describe or reference the field measurements that were performed to assess the distribution of Aedes sollicitans so that the statement here can be supported with data.

- Line 312: For Figure 8, only 12 tablet drops were reported. This is an unacceptably low sample size for a study designed measure the accuracy of an instrument, particularly since the study is simple to conduct (one tablet drop constitutes nearly 8 % of the total sample size).

- Line 313: A description of ES could not be located in the manuscript.

- Line 314 – 316: Authors state that wind had no effect on accuracy of tablet drop. However, one quarter of drop area did not contain tablets (270 – 360 degrees). This should be taken into consideration. Authors should indicate the wind direction on the figure to support that conclusion.

- Line 318: R-squared values are reported, but the statistical software used to calculate it is not described in the manuscript.

- Line 325 – 326: The conclusion that the accuracy of the tablet dropper improved as flights progressed because the UAS had more time to collect and correct GPS data is not supported by data in the manuscript.

- Line 356: The conclusion that releasing granules from 9 m was negatively impacted wind is not supported by the data as wind speed was not reported for this study and a reference indicating the wind speed that negatively affects granular drift is not provided.

- Line 371: The authors may want to also indicate the maximum application rate as vector control workers often use the maximum rate to reduce the potential for insecticide resistance to develop in mosquitoes.

- Figure 11: This figure does not add a great deal to the study, and should be considered for removal. If the authors elect to retain this figure, they should describe how the predicted flight coverage (i.e. overlap analysis) was modeled.

- Line 382 – 384: The authors should provide data to support the claim that a newly designed granular spreader can treat 1.1 ha or remove the statement.

- Lines 394 – 396: Indicating “unpublished data” appears to not be permitted by PLOS ONE.

- Line 399 – 400: Please provide the formula for calculating the estimated flow rate for VectoBac G in the RMAX.

- Line 402 – 403: Speculating on the manufacturing plans for pesticide companies may be beyond the scope of this manuscript. However, using the data from this study to encourage the manufacturers to prepare formulations for use in UAS would be well within the scope.

- Line 469: The statement that swarm technology will render payload capacity of UAS irrelevant is not supported by the data. Swam technology is likely to improve pesticide applications, but payload will likely remain an issue that will limit the duration of any application mission.

- Line 474 - 475: Please provide a reference supporting the statement that automated pesticide filling stations will be developed. If a reference is not available, it would be appropriate to encourage such development.

- Line 499 – 509: This section is overly lengthy and much is outside the scope of the study.

- Line 513: The reference Kimball & Reynolds 2016 is from a trade magazine that is not peer reviewed. There are peer-reviewed articles from the prior one or two years that may be more appropriate of a reference.

- Line 514: Please provide a reference for this statement.

- Several figures do not indicate whether the error is SEM or SD.

- A description of the statistics used in the study is not provided in the manuscript.

Reviewer #2: The paper entitled “The development of autonomous unmanned aircraft systems for mosquito control” by Williams et al. examines the delivery of insecticides for the control of larval and adult mosquitoes using modules developed for a UAS (drone) system. The paper provides a succinct history of unmanned aircraft use for mosquito control, discusses four modifications mostly made by adapting existing technologies with 3D printing technology, and provides data related to the distribution and (where relevant) droplet size for products applied using different altitudes, flying speeds and other parameters during the operation of the UAS.

The paper is well-written and provides an adequate quantitative summary and analysis of the results. I could not find any glaring shortcomings in the paper. The eleven figures seem appropriate. The references cited include many references that could be categorized as “gray literature” but they seem relevant to the text. The benefits and drawbacks of UAS technology relative to existing technologies are discussed and are summarized nicely. Given the specifics of the application results and mosquito control materials that are discussed, the paper might be more appropriate for a journal that focuses on mosquito control. Nevertheless, related papers have been published in PLoS ONE. I found this one to be an enjoyable read.

Minor criticisms are mentioned here:

Line 80: “…preferable to…”

Line 117: “Data are…”

Line 144: “alleged” flow rates? The connotation of alleged is “without proof, to have taken place or to have a specified illegal or undesirable quality.” The authors presumably do not mean the latter. Is the meaning of this statement to be that the flow rates were not confirmed or ground-truthed?

Line 167: “…servomechansim…” This should be servomechanism?

Line 294: Coquillett

Line 536: “…microhabitats…”

Line 612: Is first letter of the surname of the first author an “I” or an “L”? (See also line 86).

Line 619: Correct format of the citation.

6. PLOS authors have the option to publish the peer review history of their article (what does this mean?). If published, this will include your full peer review and any attached files.

Reviewer #1: No

Reviewer #2: No

---

## [Author Response · Author response to Decision Letter 0]

8 May 2020

Please see "Response to Reviewers" file.

---

## [Decision Letter · Decision Letter 1]

25 May 2020

PONE-D-20-02700R1

The development of autonomous unmanned aircraft systems for mosquito control

PLOS ONE

Dear Dr. Williams,

Thank you for submitting your manuscript to PLOS ONE. After careful consideration, we feel that it has merit but does not fully meet PLOS ONE’s publication criteria as it currently stands. Therefore, we invite you to submit a revised version of the manuscript that addresses the points raised during the review process.

We look forward to receiving your revised manuscript.

Kind regards,

Ahmed Ibrahim Hasaballah

Academic Editor

PLOS ONE

Reviewers' comments:

Reviewer's Responses to Questions

**Comments to the Author**

1. If the authors have adequately addressed your comments raised in a previous round of review and you feel that this manuscript is now acceptable for publication, you may indicate that here to bypass the “Comments to the Author” section, enter your conflict of interest statement in the “Confidential to Editor” section, and submit your "Accept" recommendation.

Reviewer #1: (No Response)

2. Is the manuscript technically sound, and do the data support the conclusions?

Reviewer #1: Yes

3. Has the statistical analysis been performed appropriately and rigorously? 

Reviewer #1: Yes

4. Have the authors made all data underlying the findings in their manuscript fully available?

Reviewer #1: No

5. Is the manuscript presented in an intelligible fashion and written in standard English?

Reviewer #1: No

6. Review Comments to the Author

Reviewer #1: This reviewer remains convinced that publishing this work without bioassay data that the authors apparently possess substantially limits the impact of the manuscript. Authors state that including bioassay data would complicate the manuscript. However, the authors provide a very lengthy discussion of data that is relatively simple to interpret and frame with the published literature (four figures with data: Fig 8 and 9: swath width / application rate; Fig 7: accuracy of dropping a briquette; Fig 5: liquid flow rate; Fig 6: droplet size / density).

Splitting the aeronautical / product deposition performance of the UAS and the bioassay data will require readers, if both are published, to flip between two publications to appreciate the value this research brings to the field. This reviewer sees that not combining the two as a missed opportunity to tell a complete story. If standard bioassay methods were used, they can be simply referenced in the methods to reduce the length and briefly compared in the manuscript with what is known for standard insecticide application equipment (via aircraft and truck).

However, if the authors remain steadfast on excluding bioassay data from this manuscript, this reviewer will respect that decision. I have provided comment below on how the manuscript must be revised to be acceptable for publication. A second revision is required because the Methods section is written in the present tense (past tense is standard) and the manuscript is overall very long considering the quantity of data presented. This affords the authors an opportunity to revise and condense other sections as well.

Major Issues

Over half of the abstract is introductory. Please condense.

Introduction is somewhat long and introduces equipment or approaches that are not considered further in the manuscript. The Introduction should be condensed.

The Methods includes product specifications that can be easily obtained from the manufacturer and do not need to be republished in a research article. This includes lines 100-105; sentence starting line on line 107; 117-122; 143-145.

The discussion of the data is very long and must be better focused and condensed. For example, over 900 words are used to discuss differences in liquid flow rate through the nozzles, swath width, and droplet density + diameter (Fig 5 and 6). These are fairly simple measurements to discuss, and the literature (academic, manufacturer, and regulatory) is fairly clear on what is effective for making an insecticide application. This is the case throughout the manuscript, so authors must examine the manuscript in its entirety to condense and anchor the writing in the data they collected.

Overly lengthy discussion of data muddies the message. Authors should determine key messages they wish to discuss, and discard what is tangential or not crucial and supportive.

Fig 1, 2, 3, and 4 should be combined into a single multi-panel figure (e.g. Fig 1A, B, C, D).

Figure 5 and 6: These figures are related and should be combined into a single figure with panels A and B.

Fig S3 , S4: These are links to YouTube videos of drones imaging mosquito larvae under water. They are not relevant to the results presented in the manuscript or derived from a peer-reviewed scientific study. Additionally, it is impossible for viewers of the video to know anything of the equipment, and thus are of no value for comparison to anything in the manuscript. This supplementary information must be removed from the manuscript.

Lines 261 – 263: Authors state the software used to program the 3D printer, but do not appear to have provided the STL files. Their response to this reviewer state “The STL files have been provided in the supplemental materials”, but they were not in the Appendix of supplementary figures (lines 647 – 655) or mentioned in the Materials section. As stated previously, these files will be needed by other researchers if they wish to attempt replicating what is presented in this manuscript. The model files must be provided by the authors and they must state in the Methods section where they can be downloaded.

Line 456 to end of manuscript: This section of the manuscript should start with a “Conclusion” subheading as it is not restricted to discussing the ULV module. It is also very long (nearly 1,000 words), especially considering there are four figures with data. This section will benefit a great deal by condensing the writing substantially and focusing on what can be concluded overall from the studies. Much of what is provided in line 461-end is, for the most part, tangentially related to the results and discussion. The authors may wish to consider instead of revising this section, end the manuscript with line 460 as it would simplify and speed the revision process. It is of course up to the authors how to revise this section, but it must be condensed substantially.

Minor Issues by Line Number

41: Speculative to state that UAS are preferred in congested environments as manned aircraft are very effective in such areas. Rephrase to “may be preferable”.

105: Irrelevant what controller is usually used with the UAS; the controller used in the study should be the focus.

138: liters is typically abbreviated with “L” not “l”.

149: state manufacturer, model of precision balance.

187: delete and append to prior sentence: (n = 3 replicates)

190: manufacturer and model of video goggles?

213-214: This is data and should be moved to Result section

215: see comment for line 187

226: delete “p” at end of line.

231: State parts were removed. Impossible for others to replicate the work without knowing. If lengthy, it can be included as a supplementary table / figure.

233- 239: Very lengthy; please condense.

275-278: Unclear if the authors made the measurements or if they are calculations. Values in parentheses are presumably label application rates; if so it should be stated.

297: Affect of changing nozzles on swath width was not measured. It is an overstatement that doing so “will”.

299: If this is in reference to data in a figure, the figure number should be stated.

311: If this is not the result of an experiment, it should be indicated as speculative with “this system should cover”.

314: Readers that are not experts in mosquito control will not understand “dip counts” as a measure of larval mosquito abundance. This needs to be explained in the methods. Averages of data that are presented in a manuscript must include error measurements. Such error values are absent here.

316-317: Describe how the reduction in pesticide use was calculated.

348-349: Data indicates that autonomous is more accurate and thus should be preferred. If authors interpret the data otherwise, this statement should be offered as an opinion and not a direct interpretation of the data.

390-394: Authors are introducing results that are not supported by data in this manuscript. Please remove.

498: Passos 2019 is a PhD thesis, which is unusual to include as a reference for a peer-reviewed journal article. After examining the thesis, the student analyzed imagery that was reported in a student design competition “IEEE CAS Student Design Competition 2017–2018 Finalist project”. While I’m sure there are examples of referencing PhD theses in manuscripts, there must be at least one peer-reviewed article in the literature that is more robust and relevant that can be referenced instead.

S2 Fig: Y-axis label is not standard. Please correct.

7. PLOS authors have the option to publish the peer review history of their article (what does this mean?). If published, this will include your full peer review and any attached files.

Reviewer #1: No

---

## [Author Response · Author response to Decision Letter 1]

8 Jun 2020

Response to Reviewers has been included in the uploaded files.

---

## [Decision Letter · Decision Letter 2]

18 Jun 2020

The development of autonomous unmanned aircraft systems for mosquito control

PONE-D-20-02700R2

Dear Dr. Williams,

We’re pleased to inform you that your manuscript has been judged scientifically suitable for publication and will be formally accepted for publication once it meets all outstanding technical requirements.

Kind regards,

Ahmed Ibrahim Hasaballah

Academic Editor

PLOS ONE

Additional Editor Comments (optional):

Reviewers' comments:

Reviewer's Responses to Questions

**Comments to the Author**

1. If the authors have adequately addressed your comments raised in a previous round of review and you feel that this manuscript is now acceptable for publication, you may indicate that here to bypass the “Comments to the Author” section, enter your conflict of interest statement in the “Confidential to Editor” section, and submit your "Accept" recommendation.

Reviewer #1: All comments have been addressed

2. Is the manuscript technically sound, and do the data support the conclusions?

Reviewer #1: Yes

3. Has the statistical analysis been performed appropriately and rigorously? 

Reviewer #1: Yes

4. Have the authors made all data underlying the findings in their manuscript fully available?

Reviewer #1: Yes

5. Is the manuscript presented in an intelligible fashion and written in standard English?

Reviewer #1: Yes

6. Review Comments to the Author

Reviewer #1: Thank you for addressing each of the points in the reviews. The manuscript is now much easier to read and appreciate.

7. PLOS authors have the option to publish the peer review history of their article (what does this mean?). If published, this will include your full peer review and any attached files.

Reviewer #1: No

---

## [Editor Report · Acceptance letter]

22 Jun 2020

PONE-D-20-02700R2

The development of autonomous unmanned aircraft systems for mosquito control

Dear Dr. Williams:

I'm pleased to inform you that your manuscript has been deemed suitable for publication in PLOS ONE. Congratulations! Your manuscript is now with our production department.

Kind regards,

on behalf of

Dr. Ahmed Ibrahim Hasaballah 

Academic Editor

PLOS ONE